# Surface Vertical Multi-Emission Laser with Distributed Bragg Reflector Feedback from CsPbI_3_ Quantum Dots

**DOI:** 10.3390/nano13101669

**Published:** 2023-05-18

**Authors:** Xueqiong Su, Yong Pan, Dongwen Gao, Jin Wang, Huimin Yu, Ruixiang Chen, Baolu Guan, Xinyu Yang, Yimeng Wang, Li Wang

**Affiliations:** 1College of Physics and Optoelectronics, Faculty of Science, Beijing University of Technology, Beijing 100124, China; 2College of Science, Xi’an University of Architecture and Technology, Xi’an 710055, China; 3Key Laboratory of Opto-Electronics Technology, Ministry of Education, Faculty of Information Technology, Beijing University of Technology, Beijing 100124, China; 4The College of Chemistry & Materials Engineering, WenZhou University, Wenzhou 325000, China; 5The School of Optical-Electrical and Computer Engineering, University of Shanghai for Science and Technology, Shanghai 200093, China

**Keywords:** CsPbI_3_ perovskite, QDs, DBR laser, multiwavelength

## Abstract

Quantum dots (QDs) laser has become an important way to solve micro-application problems in many fields. However, single wavelength distributed Bragg reflector (DBR) has many limitations in practical applications, such as signal transmission. How to realize multiwavelength DBR lasing output simply is a challenge. To achieve a stable multi-wavelength quantum dots laser in the near-infrared region, the perovskite CsPbI_3_ QDs laser with DBR structure is developed in this paper. A tetragonal crystal structure with complete bonding information and no defect is explained by X-ray diffractions (XRD) and Raman spectrum. The cross-section morphology of the DBR laser and the surface morphology of QDs is measured by scanning electron microscope (SEM) and transmission electron microscope (TEM), respectively. An elliptical light propagation field and a double wavelength laser radiation are obtained from the finite-difference time-domain (FDTD) simulation. The output of the three wavelength lasers at 770 nm, 823 nm, and 873 nm is measured. The emission time of a DBR laser is about 2 h, and the average fluorescence quantum yield is 60%. The cavity length selection and energy level model are put in place to clearly see the working mechanism. All the results suggest that an effective and stable CsPbI_3_ quantum dots DBR laser is realized.

## 1. Introduction

Quantum dots laser, as an important new member of semiconductor laser, has attracted much attention in recent years due to its lower threshold current density, higher quantum efficiency, narrower spectral linewidth, higher differential gain, and better temperature stability [1,2,3]. QDs laser has made continuous breakthroughs in low-threshold current and high-output power at room temperature [4,5]. Much research is devoted to the development of quantum dots with different materials to improve the laser output performance, such as graphene [6], ZnS/Se [6,7], metals [8], Si-based [9], InAs [10], ZnCdS/CdSe [11], CsPbBr_3_ [12], and mixed quantum dots [13]. Additionally, it is also a hot point to increase the laser oscillation through specific microstructure to obtain high-quality nano-laser output, such as grating [14], nano-arrays [15], liquid-core fiber [11], micro balloon sphere [12], and quantum well [16]. However, there are still some problems that must be resolved regarding quantum dots lasers, including materials and structures, how to select the appropriate quantum dots materials for improving the photofluorescence emission in a larger range and to more accurately control the doping ions, and then how to optimize the structural design of a quantum dots laser to make it conducive to the optical gain of quantum dots materials.

In a material system, all inorganic perovskite quantum dots (CsPbX_3_, X = Cl, Br, I) were synthesized by proteescu [17] in the year of 2015. This material has the advantages of high extinction coefficient, long carrier lifetime, balanced carrier mobility, low defect concentration, shallow defect energy level, high fluorescence quantum yield, and low exciton binding energy. As the size of QDs tends to the Bohr radius of the exciton, it has a more significant quantum confinement effect. Inorganic perovskite quantum dots not only have extremely high fluorescence quantum efficiency (up to 90%) and narrow emission line width (12 to 40 nm, corresponding to 410 to 700 nm emission peak), but also the emission wavelength can be conveniently and continuously adjusted by changing the elemental composition without changing the size of the quantum dots. Thus, perovskite QDs are suitable for use as the amplifying medium of nano-lasers.

The optical properties of the laser gain medium cannot be exerted without an efficient resonator structure. The optimization of laser cavity and device structure plays an important part in laser threshold, output efficiency, and stability. The distributed Bragg reflector (DBR) is an excellent reflection structure of vertical cavity surface emitting lasers (VCSEL) [18,19]. When the light is emitted from the optically sparse medium N_1_ to the optically dense medium of N_2_ (refractive index N_2_ > N_1_), the reflected light will undergo a 180° phase transition at the interface. However, the phase transition does not occur when the light is emitted from the dense light medium to the light sparse medium. When the light passes through a thin film, it will be reflected twice on the upper and lower surfaces. The thickness of the film will affect the optical path difference of the two reflections. If the thickness of the film is controlled to be a (1/4 + n) time of the wavelength, the optical path difference of the two reflections is (1/2 + 2n). The optical path difference corresponds to the p phase transition, and the two reflected lights will be in a phase superimposed and enhanced eventually. Namely, the overall reflection coefficient is raised. The DBR is actually two kinds of the medium alternately stacked of a refractive index; with light through the DBR [20], each layer will increase a certain reflection system, and the final DBR reflection coefficient can reach a very high level.

In this study, a quantum dots laser with perovskite CsPbI_3_ as the laser gain medium in a DBR reflection resonance structure is developed. We try to achieve a multi-emission laser in the near infrared region. This work starts from the measurement of a microstructure, and Raman spectroscopy is used to study the vibrational information of the element bonding. The DBR quantum dots lasers are characterized, and its reflectivity is measured. The physical model simulation method is used to calculate the optical field in the DBR laser, which is verified by the PL spectrum. The three longitudinal-mode lasers in the near-infrared region are confirmed eventually. A cavity length selection theory is proposed to illustrate the different laser emissions.

## 2. Experimental

### 2.1. The Synthetic for CsPbI_3_ QDs

The hot injection method was chosen to obtain CsPbI_3_. Caesium carbonate, octadecene, and oleic acid were placed in a three-necked flask. Heating was performed under an argon atmosphere for 60 min. Then, the temperature was increased to 160°, and heating was continued for 30 min. Forebody liquid was obtained. Next, PbI and octadecene were placed in a three-necked flask. It was also heated in an argon atmosphere at a temperature of 130°, and oleylamine and oleic acid were added after heating for 40 min. Finally, 0.4 mL of the precursor solution was quickly injected with a pipette, and 5 s later, the flask was placed in an ice-water bath for rapid cooling.

### 2.2. Preparation Process of a DBR Laser

The process of fabricating a DBR laser is shown in Figure 1. All layers of reflection were obtained by magnetron sputtering, the steps of one film: placement–feeding–background extraction–heating–pre-sputtering–sputtering–intermittent film extraction (Figure 1a,b). Vacuum degree in the sputtering cavity was pumped to $2.5\times 10^{-4}$ Pa, the temperature was heated to 100 °C, the pre-sputtering time was set to 300 s, and the number of cycles was set to 5 times. The SiO_2_ layer was deposited first according to refraction index and reflection index with 300 W, 90 sccm of Ar gas flow, and 1133 s of process time (Figure 1c). The detail of this step can be seen in Appendix A. Then, TiO_2_ was deposited on the SiO_2_ to form reflectivity and refractive index difference with 200 W, 60 sccm of Ar gas flow, and 1792 s of processing time (Figure 1d). The combination of a SiO_2_ layer and a TiO_2_ layer is called a reflector layer. After the sputtering of six layers of the DBR, the upper reflecting layers were completed (Figure 1e). Next, the perovskite CsPbI_3_ QDs were dropped on the surface of 12 layers of the DBR (The number of the DBR layers in Figure 1 is only a diagram and does not represent the real number of layers). The QDs preparation process video is provided, which can be seen in Appendix A. In the same way, the upper reflector was prepared with the same materials (SiO_2_/TiO_2_). An important step is the combination of the two 12-layer DBRs. The two DBRs were stacked and fixed directly by a clip because the quantum dots had a certain viscosity, as shown in Figure 1f–i. Therefore, the lasers of perovskite CsPbI_3_ QDs had been achieved. The picture of a DBR laser is attached to the Appendix A.

### 2.3. Characterization

Structural information of perovskite CsPbI_3_ QDs was expressed by X-ray diffraction XRD (Bruker D8 Advance, Karlsruhe, Germany ) and Raman spectra (Horiba JobinYvon, T6400, Pairs, France). The simulation of materials structure was calculated by Siesta 4.1b4. The morphology of the nanostructure was visualized by a scanning electron microscope SEM (JEOL JSM 6500F, Tokyo, Japan ). The morphology and the size were visualized by a transmission electron microscope TEM (JEM 2100, New York, NY, USA). The photoluminescence PL spectra were measured by the spectrograph (NIR512 and S2000, New York, NY, USA). The optical simulation was conducted by finite-difference time-domain.

## 3. Result and Discussion

Information of microstructures for CsPbI_3_ is demonstrated in Figure 2a. The crystal structure is confirmed at this measurement. The peaks corresponding to crystalline plane of (023), (015), (122), and (016) are marked according to the standard card (JCPDS 37-1463), which suggest a tetragonal-type structure in CsPbI_3_ QDs. The characteristic symmetry of the four high-order axes determines that the sum of the two paraxial basis vectors of the tetragonal crystal system must be perpendicular to the principal axis, equal in size, and orthogonal to each other; that is, the unit cell must have the shape of a square column, and the unit cell parameter is a = b ≠ c, α = β = γ = 90°. The single diffraction peak of cationic Cs^+^ is not found, indicating that it is completely doped into the quantum dot. More specific microscopic information needs to be tested because the results in XRD are not enough to fully explain. Thus, the Raman spectrum is conducted to understand the bonding information, as shown in Figure 2b. Raman peaks are mainly concentrated in the short-wave region. The peaks at 214 and 316 cm^−1^ can be explained by the bond vibration of CsPbI_3_, which confirms that Cs was completely incorporated [21]. The surface scattering phonon signal of the tetragonal structure can be observed and reveals peaks at 621 and 860 cm^−1^, confirming the integrity of the quantum dots perovskite structure and enabling full performance from its physical properties [22,23]. The bonding information of Pb-I is also found according to the peak of 1130 cm^−1^ [24,25]. Their vibration signals are weak because of the presence of Pb and me inside the structure. Finally, the Raman vibration peak representing the quartz substrate is 1372 cm^−1^. The reason for the quartz substrate peak is that we coated quantum dots on a quartz substrate for Raman measurement.

The cross-section SEM images of the DBR laser are displayed in Figure 3a–c. The compact and seamless cross-section morphology is shown in Figure 3a, which illustrates that the preparation of the DBR meets the expected requirements. The enlarged cross-section area is shown in Figure 3b, which reveals the cross-section morphology of SiO_2_ and TiO_2._ Among them, SiO_2_ is represented by a black stripe area, and TiO_2_ is represented by a white stripe area. There are no obvious defects and gaps on the surface between each of the two layers. Furthermore, the thickness of SiO_2_ and TiO_2_ is marked (Figure 3c). The average thickness of SiO_2_ is 126 nm, which is larger than 78 nm corresponding to TiO_2_. The total thickness of the 12 DBR layers is about 1224 nm (the thickness difference of each layer is less than 5 nm). The change in thickness between the two surfaces will directly result in the alteration of reflection and refraction effecting when the light oscillates in it. The top surface of the DBR is also characterized. There are still many impurities on the surface caused by dust adsorption, according to Figure 3d. Figure 3e,f exhibits the surface morphologies of the perovskite quantum dots via TEM images in different scales. Regular square-shape uniform distribution of perovskite quantum dots is presented. The size distribution of QDs ranging from 11 to 25 nm is confirmed according to TEM image and sampling statistical method. The emission color changed from 730 nm to 900 nm. However, about 60% of quantum dots have a size of 17 nm. Thus, the spectrum is mainly concentrated on one color, which produces oscillations that suppress the formation of laser light from quantum dots of other sizes (mode competition). Their dispersion is good, and the cluster phenomenon is not obvious. Thus, they can be regarded as evenly distributed in the DBR gain dielectric layer.

Figure 4 describes the reflectivity test on both sides of the DBR. A complete DBR laser is used for each test. This measurement can be used to predict and analyze the absorption and oscillation of light in the DBR. The oscillation characteristics are shown by the reflectivity test results of the top reflector. The average reflectivity in the range of 300–625 nm is only 19.8%, the highest reflectivity is 36.7%, and the reflectivity at 355 nm is 23% (Figure 4a). These data indicate that the excited state is larger than the irradiated state before the 600 nm wavelength. Since the total reflectance is lower, more absorption of photons is confirmed. In this experiment, a puled light with a wavelength of 355 nm from the Nd:YAG pulsed laser is selected as the excitation light, and a reasonable range of its absorption characteristics in the DBR is demonstrated. In addition, the reflectivity increases sharply after the wavelength of 625 nm. This indicates that the DBR is likely to radiate in this band, especially in the range of 700–900 nm. Reflectivity of 99.8% at 700 nm, 94% at 770 nm, 81.3% at 823 nm, and 61.2% at 873 nm is confirmed. The test results of the bottom reflector (Figure 4b) show the same trend as those of the top reflector. This indicates that the influence of the trans-deep UV substrate on the incident light and on the reflected light can be neglected. The effective design and preparation of the DBR is indirectly shown by the reflectivity measurement. If the number of layers is too small, the emissivity of the booster cavity will be low, and it will not be able to form an effective resonance. At the same time, if the number of layers is too large, the reflectivity will not increase because of more losses.

The DRR model is established, and then the simulation is performed using the finite-difference time-domain (FDTD) method. The details can be seen in the Appendix A. In the simulation, the perovskite CsPbI_3_ QDs are randomly arranged, and the number and thickness of the reflective layers are consistent with the experimental data. The simulation results show the multilongitudinal-mode optical radiation generated by the DBR laser. The output longitudinal mode is observed separately by setting the filter. Among them, the dynamic process of the 750 nm emission in the view of the XY plane is shown in Figure 5a–d. First, the optical oscillator is generated when the incident light enters the DBR reflector. Currently, the CsPbI3 QDs start to absorb the photon energy but do not produce any radiation (Figure 5a). Subsequently, the energy accumulation causes the colony number to invert, producing radiation in the 750 nm mode, but with a low intensity (Figure 5b). After that, the 750 nm is oscillating in the DBR for mode competition (Figure 5c). Finally, the 750 nm mode is enhanced and radiated as the accumulation of the oscillation with the TEM10 transverse mode (Figure 5d). In the meantime, the 833nm mode is recognized. The dynamic process of 833 nm emission in XY plane is shown in Figure 5e–h, which is the same with 750 nm mode. However, the intensity of 833 nm mode is weaker than that of 750 nm mode, and the transverse mode (TEM11) is also different. The reason for this change can be explained by different energy levels in the material absorbing energy and emitting different photon shapes. In order to clearly understand the dynamic process of multilongitudinal-mode light output, the light evolution process in the XZ cross-section plane is also studied, as shown in Figure 5i–l. An elliptical propagation oscillation band is formed around the middle-layer coated CsPbI_3_ QDs gain medium after the incident light enters the DBR, which is caused by the fixed-difference gradient between the reflectivity and refractive index of each layer (Figure 5i). There should be many elliptical propagations and bands, not only the one in the whole DBR laser because the range of simulation is built within 150 nm. The propagation band strength of the ellipse gradually decreases as the photon absorption of the quantum dots increases. At the same time, the elliptical propagation band is symmetrically divided through many black stripes. These black stripes can be interpreted as the generation of weak multimode radiation (Figure 5j). The energy that is absorbed by perovskite continuously accumulates and radiates a stronger output, resulting in a more obvious division to the elliptical propagation band. The vertical emission of the multimode laser has been generated on the DBR surface (the top of Figure 5k). We have reason to believe that this is not the reflected light. The reflected light can be produced at the initial stage according to Figure 4. Ultimately, the output light oscillates stably after the mode competition, which makes the reflective layer from the stripe oscillation field strength. The vertical emission intensity on the surface of the DBR reaches the maximum. Therefore, the multilongitudinal laser is the output.

The absorption and PL emission of CsPbI_3_ QDs is presented in Figure 6a. The absorption at 660 nm was confirmed, which differs from the Stokes shift of the fluorescence peak at 172 nm. Fluorescence coverage at 700–900 nm was confirmed, which covered the laser emitting. Meanwhile, the fluorescence lifetime reached at 18 ns, as shown in Figure 6b. Under the prediction of the model and simulation study, the photoluminescence spectrum is measured at room temperature. The laser can not excited the materials, as the pump power density is in the range of 0.6–1.6 MW/cm^2^. When the incident increases to 1.9 MW/cm^2^, the multilongitudinal-mode laser output is generated. Therefore, 1.9 MW/cm^2^ can be considered as the threshold of this CsPbI_3_ QDs DBR laser. The wavelengths of three longitudinal modes are 770 nm, 823 nm, and 873 nm, respectively, which is given as the $\nu_{1},\nu_{2}$, and $\nu_{3}$. There are only two kinds of longitudinal-mode wavelength optical output in FDTD simulation, including 750 and 833 nm. The results in the PL spectrum close to the simulation are presented, except the third longitudinal output. The laser output of the three modes is continuously enhanced with the increase in the power density of the excitation light. The maximum pump power is set at 3.1 MW/cm^−2^. Among the three modes, the intensity of $\nu_{1}$ is higher than that of the other two modes of the same pump power density. The reason for this phenomenon can be attributed to the imbalance of energy absorption caused by the differences of in structure of the CsPbI_3_ QDs. Close frequency intervals of $\nu_{1},\nu_{2}$, and $\nu_{3}$ are calculated, which is the output characteristic of the composite FP cavity. Therefore, near-infrared nano-laser output is confirmed. The photon lifetime at 770 nm wavelengths beyond the threshold is shown in Figure 6b, which illustrates that above the threshold (1.9 MW/cm^−2^), SE is dominated with a single exponential decay time of 11 ps. Compared with the photon lifetime below the threshold, the photon lifetime decreases by 804 ps, indicating that the fluorescence residence time is shortened after laser generation, and the energy level corresponds to the dominant SE mode. The linewidth is larger before reaching the threshold and smaller after the threshold, which is consistent with the rule of the laser excitation process (Figure 6c). The average linewidth after the threshold is 1.6, 2.3, and 2.8 nm, corresponding to the wavelengths of 770, 823, and 873 nm, respectively. The relationship between peak intensity and pump power density is also shown in Figure 6d. The linewidth narrows as the peak intensity increases. The average peak intensity at $\nu_{1}$-770 nm is higher than that of the other two modes, which is consistent with the laser spectrum. The measurement results of linewidth and peak intensity show that the obtained nano-laser can be applied in practice (Figure 6e,f).

Figure 7a shows the relationship between laser intensity and time. At 40 min, the laser can maintain a good output intensity above 70%. The output intensity gradually decreases after 60 min. Finally, the output is terminated in 117 min. Excitation conditions are kept constant, and stability lies around 5 %. This indicates that the continuous output time of our DBR device is about two hours. Meanwhile, the quantum yield (QY) is also on display. In Figure 7b, the relationship between the QY and the QDs solution concentration is plotted and fitted. When the concentration of the QDs solution is 5%, the quantum yield is about 25–40%. The quantum yield is stable between 58 and 66% when the concentration is gradually increased. The average quantum yield of the perovskite quantum dots synthesized in this work is 60%.

The emission process of the multiwavelength laser is shown in Figure 8a. Optical oscillation occurs when the incident light is perpendicular to the DBR surface. The interlayer oscillation mode is the FP type. The light is amplified in the upper and lower reflectors after passing through the CsPbI_3_ QDs layer. After absorbing energy, the QDs emit photons. These photons produce laser radiation due to population inversion. In this process, there must be mode competition. Then, the laser with a specific wavelength and frequency will be generated at the output. Furthermore, the laser emission peak and fluorescence peak are compared in Figure 8b. The 770 nm and 823 nm laser peaks are located to the left of the central wavelength of the 832 nm PL peak, which can be classified as a type I emission [26]. This type of laser is caused by the high reflectivity of the DBR (94% of 770 nm, 81.3% of 823 nm). At the same time, 873 nm is located to the right of the PL center, which is recognized as type II in the two-photon absorption [27]. When the low-reflectivity DBR is not enough to complete the laser radiation, the two-photon absorption becomes the main way of gain in the cavity. Although the quantum dots prepared in this paper are not a core–shell structure, they are in the DBR reflection structure, so they can still meet the requirements of the two-photon absorption [26,27,28]. Here, three longitudinal modes of $\nu_{1},\nu_{2}$, and $\nu_{3}$ are proved (Figure 8c). The wavelength or frequency of the different output lasers is directly related to the length of the oscillation cavity, which can be expressed by the formula below [29,30,31]:(1)$$L=k\frac{\lambda_{k}}{2}k=1,2,3$$
where *L* is the length of cavity, *k* is the coefficient, and $\lambda_{k}$ is the oscillation wavelength. Only when this condition is satisfied will the laser light oscillate. Thus, the longitudinal mode can be calculated by the following formula [29,30,31]:(2)$$\nu_{k}=\frac{c}{\lambda_{k}}=k\frac{c}{2nL}$$
where $\nu_{k}$ is the longitudinal-mode frequency, c is the speed of light, and *n* is the effective refractive index. Therefore, the wavelength is proportional to the cavity length. According to the experimental data and the cavity-length selection formula, the cavity lengths corresponding to the three frequencies in the DBR are calculated, which are 1549 nm, 1646 nm, and 1748 nm, respectively. The choice of cavity length is dependent on the total length of the DBR. The three frequencies of light radiation correspond to the different photon transitions in the material. It is reasonable to assume that the absorbed energy is 3.496 eV due to the excitation of the 355 nm pulsed laser. According to the microstructures of perovskite, the energy level positions of the ground and excited states of the materials with the type I addition are not different from those of traditional perovskite materials. The energy position of −5.97 eV is the energy position of the ground state, and −3.65 eV is considered to be the energy position of the excited state. The absorption and emission of the photons between the valence band maximum (VBM) and conduction band minimum (CBM) are assumed because of their direct band gap properties. The gain decay and the volume fraction of the perovskite can be expressed by the formula below according to the Refs [32,33]:(3)$$\tau_{s}=\frac{4\pi R^{3}}{3}\frac{n_{r}}{\xi \sigma_{g}c}$$
(4)$$\xi \sigma_{g}>\frac{4\pi n_{r}}{3c\beta }$$
(5)$$\xi =\frac{4\pi n_{0}R^{3}}{3}$$
where the $\tau_{s}$ is the stimulated emission buildup time, *R* is the size of QDs, $n_{r}$ is the sample refractive index, $\sigma_{g}$ is the gain cross-section, and *c* is the light velocity, $\beta \sim 5\;ps\;nm^{-3}$. The above formula can be used to calculate and measure the microscopic absorption and fluorescence emission of perovskite quantum dots. As the wavelength increases, the position of the radiation energy level decreases as the wavelength increases due to the decreasing radiation energy. After absorbing the photon energy and oscillating, the photon radiation is produced. The corresponding radiation energies of the three modes are 1.611, 1.506, and 1.420 eV, respectively (Figure 8d).

## 4. Conclusions

In this study, a CsPbI_3_ QDs DBR laser with three longitudinal modes in the near-infrared wavelength region is presented. The microstructure information of CsPbI_3_ QDs is investigated by XRD and Raman spectrum, which suggests a crystal structure with complete bonding information. The interfacial morphology of the DBR laser and the surface morphology of the QDs were characterized by SEM and TEM, respectively. The reflectivity measurement indirectly illustrates the effective working structure of the prepared DBR laser. The light emission and laser output in the DBR laser are computed using FDTD, which reveals the elliptic light emission field and double longitudinal-mode laser radiation. Finally, the feasibility of the perovskite DBR laser is demonstrated by measuring the output of the three longitudinal-mode lasers at 770, 823, and 873 nm by near-infrared PL spectrum. Thus, the cavity length selection and the energy level model are established to explain the principle of generating the three longitudinal-mode lasers. In conclusion, we have prepared inorganic perovskite quantum dots suitable for laser gain medium and demonstrated their optical anisotropy. Simultaneously, the DBR structure quantum dots laser combining this gain medium is realized, demonstrating three longitudinal-mode laser output. This is a new report on inorganic perovskite quantum dots laser radiation in the near-infrared region.

## Figures and Tables

**Figure 1 nanomaterials-13-01669-f001:**
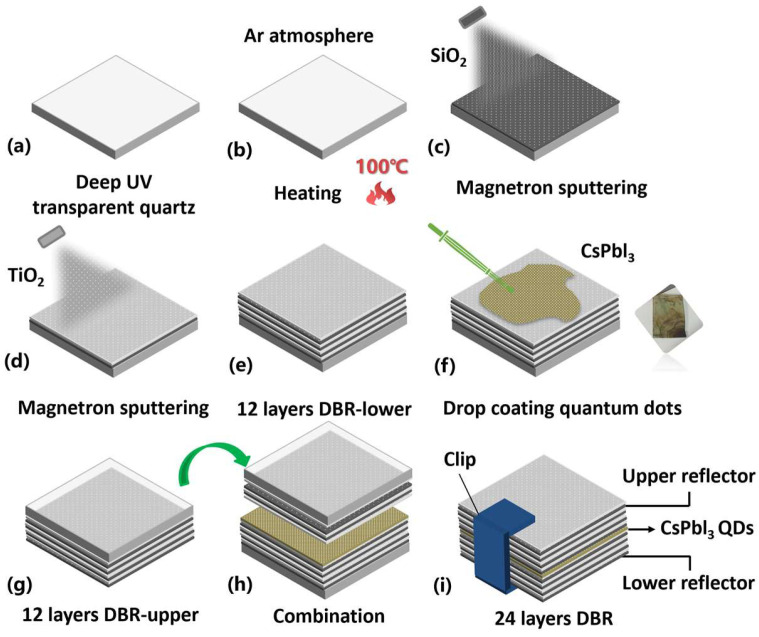
Preparation process of a perovskite QDs DBR laser: (**a**) Selection and cleaning of substrate; (**b**) heating treatment of substrate in Ar gas atmosphere; (**c**) SiO_2_ thin-film deposition by magnetron sputtered; (**d**) TiO_2_ thin-film deposition by magnetron sputtered; (**e**) 12 layers of the DBR lower deposition; (**f**) drop coating CsPbI_3_ quantum dots on the surface; (**g**) 12 layers of the DBR upper deposition; (**h**) the physical combination of the upper and lower layers of the DBR; (**i**) fixation of upper and lower layers of the DBR.

**Figure 2 nanomaterials-13-01669-f002:**
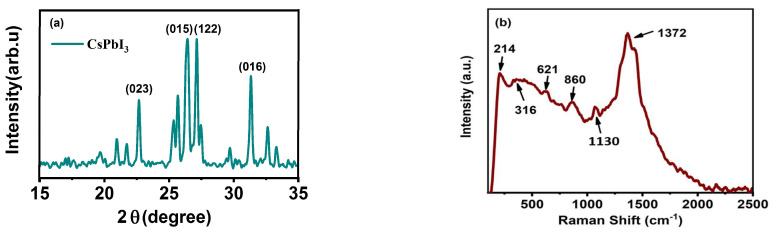
The XRD patterns and Raman spectrum of perovskite CsPbI_3_ QDs: (**a**) The XRD patterns of CsPbI_3_ QDs; (**b**) the Raman spectrum of perovskite CsPbI_3_ QDs.

**Figure 3 nanomaterials-13-01669-f003:**
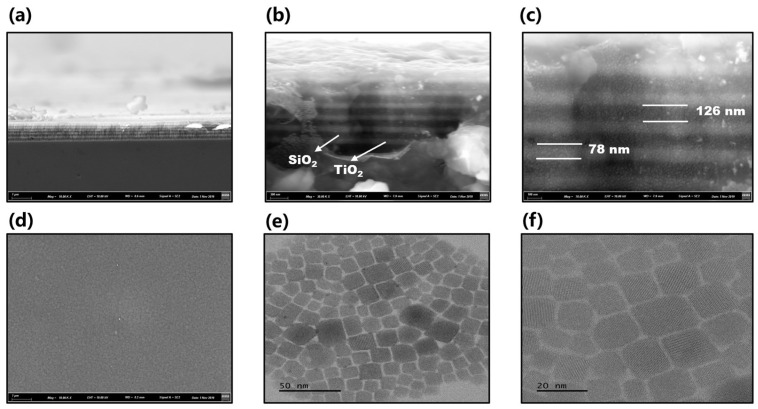
The SEM images of the DBR laser and TEM images of perovskite CsPbI_3_ QDs.: (**a**) The SEM image of compact and seamless cross-section morphology of the DBR; (**b**) the SEM image of enlarged cross-section area of the DBR; (**c**) the SEM image of the thickness of SiO_2_ and TiO_2_; (**d**) the SEM image of the surface on the DBR; (**e**,**f**) the TEM image of morphologies of perovskite quantum dots.

**Figure 4 nanomaterials-13-01669-f004:**
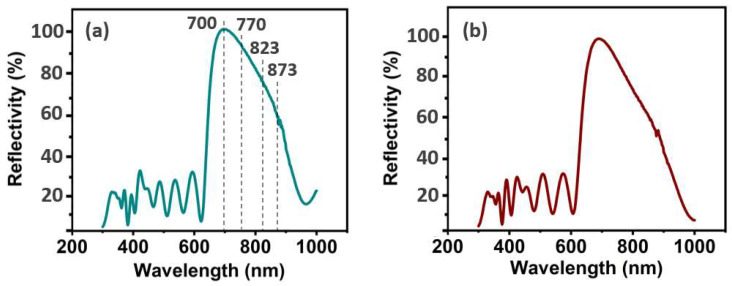
The reflectivity on the surface of the upper and lower reflector in the DBR laser: (**a**) The reflectivity on the surface of upper reflector in the DBR; (**b**) the reflectivity on the surface of lower reflector in the DBR.

**Figure 5 nanomaterials-13-01669-f005:**
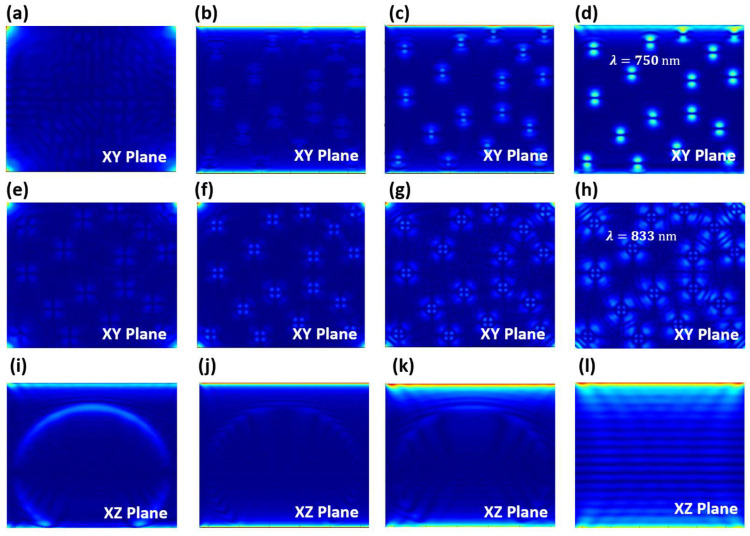
The FDTD simulation of the DBR laser: (**a**–**d**) The dynamic process of 750 nm emission detected in XY plane; (**e**–**h**) the dynamic process of 833 nm emission detected in XY plane; (**i**–**l**) the dynamic process of multilongitudinal-mode emissions in XZ plane.

**Figure 6 nanomaterials-13-01669-f006:**
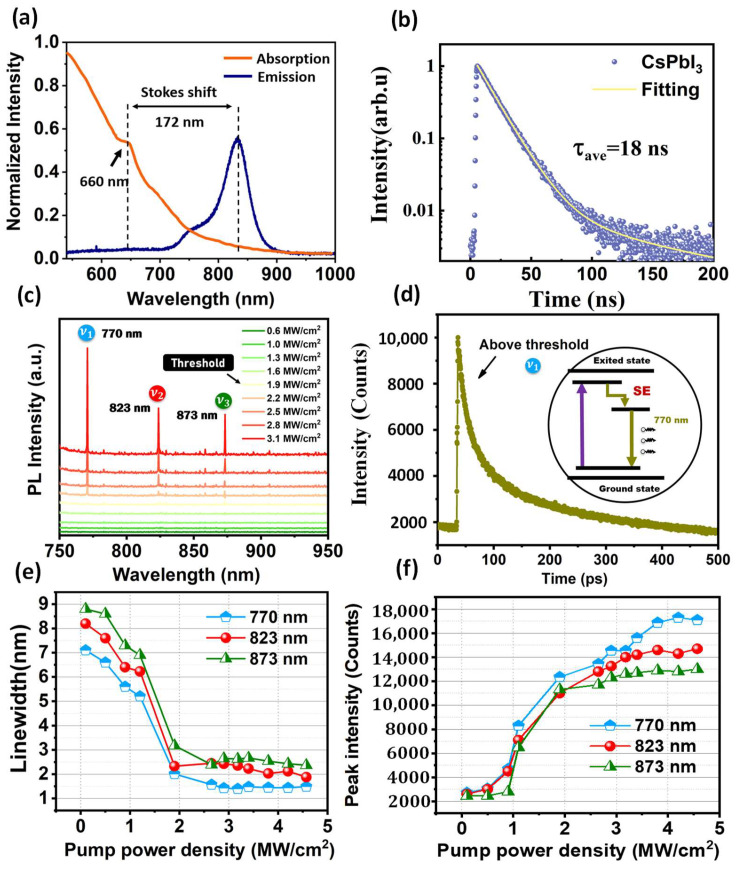
The PL spectrum of perovskite CsPbI_3_ QDs and DBR laser: (**a**) The absorption and emission of CsPbI_3_ QDs; (**b**) fluorescence lifetime of CsPbI_3_ QDs; (**c**) multilongitudinal-mode laser behavior at room temperature; (**d**) decay times measured at pump intensities above threshold in 770 nm; (**e**) the relationship between power density and linewidth; (**f**) the relationship between power density and peak intensity.

**Figure 7 nanomaterials-13-01669-f007:**
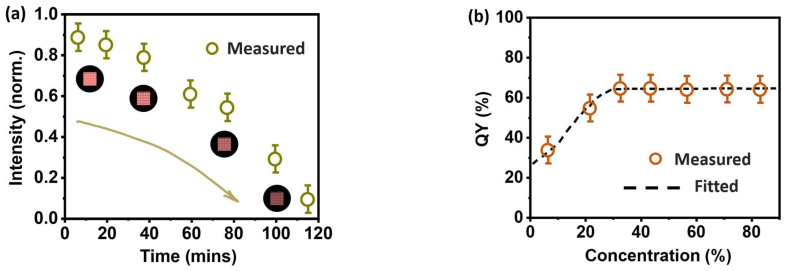
The duration and quantum yield of the QDs device: (**a**) The relationship between lasing intensity and the time duration. Inset: the change in the lasing picture with time; (**b**) the QY of the QDs device with different concentration.

**Figure 8 nanomaterials-13-01669-f008:**
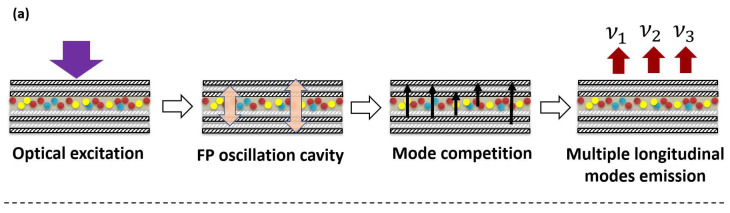
The physical mechanism involved in the research: (**a**) Multiwavelength mode output; (**b**) the gain mechanism of two photon absorption in QDs; (**c**) relationship between cavity length and wavelength; (**d**) energy level of radiation wavelength.

## Data Availability

Data is contained within the article or Appendix A.

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
