# Peer review of "Surface Vertical Multi-Emission Laser with Distributed Bragg Reflector Feedback from CsPbI3 Quantum Dots"

_nanomaterials, 2023, doi:10.3390/nano13101669_

Round 1

Reviewer 1 Report

The manuscript entitled “Surface vertical multi-emission laser with DBR feedback from CsPbI3 quantum dot” was interesting and well written by the authors. The concept of multi-wavelength quantum dot laser is impressive and all the necessary characterizations were done and the discussion has been supported by the experimental data. However, the authors needs to address some minor issues before its publication

1. The reflectivity properties were presented in Figure 4, but in the manuscript the authors mentioned it as Figure 3, which needs to be corrected.

2. The authors did not mentioned anything about size of the CsPbI3 QDs and from the TEM image presented in the manuscript, the size of the QDs looks uneven. Generally, the size of the QDs play an important role in varying the emission color. But in the present case, the quantum confinement effect have nothing to do with multi-wavelength QD laser?   

Author Response

Thank you very much for giving the manuscript entitled “Surface vertical multi-emission laser with DBR feedback from CsPbI3 quantum dot” (Manuscript ID: 2323960) the chance for revision.

We appreciate the valuable comments of editor and reviewers very much. We have studied comments carefully and have made correction which we hope meet with approval. We listed our revisions below and used Orange-colored in text.

We hope that we have addressed the reviewers’ concerns and the manuscript is now suitable for publication in Nanomaterials.

Thank you very much!

Reviewer #1:

  1. The reflectivity properties were presented in Figure 4, but in the manuscript the authors mentioned it as Figure 3, which needs to be corrected.

Response:

We are very sorry for this negligence.

The figure 3 and figure 4 have been corrected.

Thank you very much.

  1. The authors did not mentioned anything about size of the CsPbI3 QDs and from the TEM image presented in the manuscript, the size of the QDs looks uneven. Generally, the size of the QDs play an important role in varying the emission color. But in the present case, the quantum confinement effect have nothing to do with multi-wavelength QD laser?

Response:

Thank you very much for your comment.

The size data of CsPbI3 QDs are added in the text according to sampling statistical method.

The size distribution of QDs ranges from 11 to 25 nm are confirmed. In fact, here is the existence of quantum confinement effect based on PL spectra. The emission color changed from 730 nm to 900 nm. However, about 60% of quantum dots have a size of 17 nm. Thus, the spectrum is mainly concentrated on one color, which produces oscillations that suppress the formation of laser light from quantum dots of other sizes (mode competition).

For clearly, this explanation has added in the text.

Once again, thank you very much for your comments and suggestions.

We tried our best to improve the manuscript and made some changes in it.

We appreciate for Editor and Reviewers’ warm work earnestly, and hope that the correction will meet with your approval.

Reviewer 2 Report

The authors are presenting their results focusing on the growth and characterization of CsPbI3 quantum dots toward Bragg reflector lasers. The paper needs improvements in several areas.

1- The captions of Figures 1 to 4 need to be expanded.

2- The authors to elaborate in detail the directions how to improve the stability and the lifetime of the device. It is not clear if these perovskite quantum dots would remain stable in the ambient air.

3- The supplementary section is like a laundry list of figures. The captions need to be expanded and text need to be included.

4- There are typos, such as in the caption of Fig. 3S, 1*1 the * needs to be replaced by a proper symbol. In Line 94 of the main text, the section title is sitting alone at the end of the page. The entire text should be proof-read carefully. Fig. 8b is also too small.

Round 2

Reviewer 2 Report

1- The authors still did not improve the supplementary document properly. The title is not even the same as the main paper. They added texts that are not even having the same font size and spacing. They need to do a proper job not in a hurry to put a proper document together.

2- The main text was not proof read properly either. In the abstract the DBR should be introduced in line 17 and not 21. Also DBR should be written in full term as distributed Bragg reflector in the title. The same thing for the title of  supplementary.

3- Line 208 the sentence highlighted in yellow referring to 355 nm wavelength is written in a confused manner.

Round 3

Reviewer 2 Report

The paper still suffers from major inconsistency. For example the term Figure in the main text sometimes referred as figure. It should be checked and pick a proper notation for the entire text.

There are grammar issues in the main text too. For example:

Page 4 of the main text first paragraph, it should be: The information of the miscrostructures for .....

The main text needs a proper proof read to catch grammar issues and typos such as the ones noted.

In the supplementary document there are still issues, for example page 3:

... release energy, The target.....

the comma after energy should be full stop.

Page 5: the following part of the sentence doesn't make sense:

FDTD materials base on library to establish the new material

and also base should be based. There should be also a library not library.

This document needs further polishing which is the job of the authors.
